# Identification of the Magna Radicular Artery Entry Foramen and Adamkiewicz System: Patient Selection for Open versus Full-Endoscopic Thoracic Spinal Decompression Surgery

**DOI:** 10.3390/jpm13020356

**Published:** 2023-02-17

**Authors:** Roth Antonio Vargas, Eduardo Miquelino De Olinveira, Marco Moscatelli, Jorge Felipe Ramírez León, Morgan P. Lorio, Rossano Kepler Fiorelli, Albert E. Telfeian, Ernest Braxton, Michael Song, Kai-Uwe Lewandrowski

**Affiliations:** 1RIWO Spine Center of Excellence, Department of Neurosurgery, Foundation Hospital Centro Médico Campinas, Campinas 13083-210, Brazil; 2Proton Diagnósticos Hospital Centro, Campinas 13083-190, Brazil; 3Clinica NeuroLife, Natal 59020-300, Brazil; 4Minimally Invasive Spine Center, Reina Sofía Clini Department of Orthopaedics, Fundación Universitaria Sanitas, Bogotá 111321, Colombia; 5Advanced Orthopedics, Altamonte Springs, FL 32701, USA; 6Department of General and Specialized Surgery, Gaffrée e Guinle University Hospital, Federal University of the State of Rio de Janeiro (UNIRIO), Rio de Janeiro 22290-250, Brazil; 7Department of Neurosurgery, Rhode Island Hospital, The Warren Alpert Medical School of Brown University, Providence, RI 02903, USA; 8Vail Summit Orthopaedics & Neurosurgery, Frisco, CO 80443, USA; 9Advanced Neurosurgery, Reno, NV 89521, USA; 10Center for Advanced Spine Care of Southern Arizona, Tucson, AZ 85712, USA; 11Department of Orthopaedics, Fundación Universitaria Sanitas, Bogotá 111321, Colombia; 12Department of Orthopedics, Hospital Universitário Gaffre e Guinle, Universidade Federal do Estado do Rio de Janeiro, Rio de Janeiro 21941-901, Brazil

**Keywords:** artery of Adamkiewicz, spinal cord blood supply, anterior spinal cord syndrome

## Abstract

Background: Casually cauterizing the radicular magna during routine thoracic discectomy may have dire consequences. Methods: We performed a retrospective observational cohort study on patients scheduled for decompression of symptomatic thoracic herniated discs and spinal stenosis who underwent a preoperative computed tomography angiography (CTA) to assess the surgical risks by anatomically defining the foraminal entry level of the magna radicularis artery into the thoracic spinal cord and its relationship to the surgical level. Results: Fifteen patients aged 58.53 ± 19.57, ranging from 31 to 89 years, with an average follow-up of 30.13 ± 13.42 months, were enrolled in this observational cohort study. The mean preoperative VAS for axial back pain was VAS of 8.53 ± 2.06 and reduced to a postoperative VAS of 1.60 ± 0.92 (*p* < 0.0001) at the final follow-up. The Adamkiewicz was most frequently found at T10/11 (15.4%), T11/12 (23.1%), and T9/10 (30.8%). There were eight patients where the painful pathology was found far from the AKA foraminal entry-level (type 1), three patients with near location (type 2), and another four patients needing decompression at the foraminal (type 3) entry-level. In five of the fifteen patients, the magna radicularis entered the spinal canal on the ventral surface of the exiting nerve root through the neuroforamen at the surgical level requiring a change of surgical strategy to prevent injury to this important contributor to the spinal cord’s blood supply. Conclusions: The authors recommend stratifying patients according to the proximity of the magna radicularis artery to the compressive pathology with CTA to assess the surgical risk with targeted thoracic discectomy methods.

## 1. Introduction

Endoscopic spine surgery is becoming more popular amongst spinal surgeons [1,2,3]. Its clinical indications have been expanded from the lumbar into the cervical spine and to a limited extent into the thoracic spine as a minimally invasive and less burdensome alternative to the more traditional laminectomy, costotransversectomy, and extracavitary approach to the anterior thoracic spine [4]. Segmental vessels are sometimes encountered during open and endoscopic surgery in the thoracic spine [5]. Depending on the surgical level, the segmental artery and vein may overlay or intrude into the disc space and, therefore, be encountered during the endoscopic decompression [5]. The surgeon may casually consider the ligation of these vessels, mainly if access to the herniated thoracic disc cannot be created by bony decompression of the facet joint or the adjacent rib head alone [6].

The authors of this article want to raise awareness of the arterial Adamkewicz system (AKA) that may be encountered during thoracic decompression surgery and offer patient stratification criteria to aid in selecting patients for modern minimally invasive including endoscopic versus open decompression surgery in the thoracic spine [7]. The location, origin, and course of the artery of Adamkiewicz—also known as the magna radicularis—varies in the thoracic spine [8]. While spinal surgeons may be well aware of potential injury during the more traditional thoracic spine operations, considering the damage to the arterial system that provides vital blood supply to the thoracic spinal cord may not necessarily be on their minds when performing endoscopic surgery. Spinal endoscopy can effectively relieve pain when performed on validated pain generators by highly skilled surgeons [9,10,11,12,13,14,15]. However, it also suffers from the stigma of a minor procedure that is not only performed by surgeons [16,17]. Nowadays, many spine care providers with a background in interventional radiology and pain management [18,19,20,21,22,23,24] without postgraduate training in spine surgery are performing these endoscopic decompressions at varying degrees of skill level and complexity. Because of this common trivialization of the procedure and much lower reported complication rates when compared to open translaminar spine surgery [25], a false sense of security and overconfidence may arise in the operating room.

However, an injury to the Adamkiewicz system may have devastating consequences and may blindsight the spinal surgeon particularly when the operation is carried out in an ambulatory surgery center without the full backup of other ancillary services and departments that would be needed in case of paraplegia. In this paper, the authors present their experience with using the magna radicularis protocol in stratifying patients for open versus endoscopic thoracic disc decompression. They present an illustrative retrospective cohort study and their preoperative protocol to work patients up thoroughly for the presence of the AKA at the surgical level or in close proximity before attempting thoracic decompression surgery.

## 2. Materials and Methods

### 2.1. Study Design

Patients were enrolled prospectively in an observational sequential cohort study after the first AKA patient was identified serendipitously (Figure 1). Between March 2018 and August 2021, patients with clinical signs of symptomatic thoracic disc herniation were treated with endoscopic decompression using a preoperative computed tomography angiography (CTA) protocol described below to determine whether patients were at risk of neurological deterioration from injury to the AKA system due to iatrogenic spinal cord ischemia. There were 10 (66.7%) female and 5 (33.3%) male patients with an average age of 58.53 ± 19.57, ranging from 31 to 89 years. The average follow-up was 30.13 ± 13.42 months. The study’s IRB approval number is CEIFUS 106-19.

### 2.2. Experimental Procedures

We performed a retrospective observational cohort study on patients scheduled for decompression of symptomatic thoracic herniated discs and spinal stenosis who underwent a preoperative computed tomography angiography (CTA) to assess the surgical risks by anatomically defining the foraminal entry level of the magna radicularis artery into the thoracic spinal cord and its relationship to the surgical level. The authors’ hypothesis was that through scientific experimental design and detailed clinical research, a patient stratification protocol could be devised according to the proximity of the magna radicularis artery to the compressive pathology as determined with CTA to assess the surgical risk with targeted endoscopic thoracic discectomy methods.

### 2.3. Patient Selection & Inclusion/Exclusion Criteria

Patients were included in the study if they presented with continuous thoracic back or flank pain. Symptoms were required to be present for at least six months or more before the consultation, despite a recommended three-month physiotherapy regimen and medical and interventional pain management. Patients were further evaluated for clinical history and physical examination findings consistent with a thoracic disc herniation, including sensory deficits, motor weakness, and the presence of any upper motor neuron signs. Necessary diagnostic imaging included standing X-rays, thoracic magnetic resonance imaging (MRI), and computed tomography (CT) scans, including CTA, to study the AKA system and its potential contribution to the segmental tributaries at the thoracic surgical level. Patients were excluded if they had any underlying neurological condition or spinal cord compression in distant parts of their spine affecting their gait cycle and locomotion. Additional exclusion criteria for endoscopic decompression were severe bony central canal or foraminal stenosis, excessive coronal and sagittal plane deformity in excess of 40 degrees, conus medularis syndrome, systemic neuropathy or spinal tumors, blood dyscrasia, pregnancy, allergies, mental handicaps, or psychiatric conditions precluding adequate communication or language problems.

### 2.4. Identification of the AKA

The initial thoracic disc herniation patient enrolled in this study had a CTA for a workup of a thoracoabdominal aortic aneurysm [26,27]. The patient’s AKA system was found to enter the spinal canal at the proposed surgical level in the thoracic spine [28]. This observation prompted the authors to evaluate each thoracic disc herniation patient with a formal CTA study to avoid neurological complications. There are several documented cases of change in surgical approach if the AKA and planned approach are on the same side [29,30,31,32]. The existing literature suggests that the AKA must be distinguished from the anterior radiculomedullary vein [33]. The latter is very similar in shape and may follow a very close course to the artery of Adamkiewicz [34]. The authors employed the “continuity technique“ to avoid confusing the radiculomedullary vein with the AKA [35]. This technique entails tracing the vessel suspected to be the AKA, back to its origin from the aorta. To better assess the surgical risk of injuring the AKA system, the relationship between its foraminal entry-level and the symptomatic compressive pathology was divided into three types: (1) Far type (the pathology is far from the AKA), (2) Near type (the pathology is not in the AKA entry foramen but in close proximity; i.e., at an adjacent level), and the (3) Foraminal type (the pathology is in the AKA entry foramen—the most dangerous type).

### 2.5. Endoscopic Surgery Technique

The authors chose a lateralized interlaminar approach to the thoracic disc herniation in 13 out of 15 cases to avoid having to deal with the rib head attachment to the costovertebral joint and to gain direct access to the thoracic intervertebral disc. The relationship between the rib head and the thoracic intervertebral disc level may vary and obstruct access to the painful compressive pathology. This miniaturized posterior muscle splitting approach also diminishes the collateral exposure-related damage significantly, but the working area is limited to the interlaminar window allowing resection of the yellow ligament and decompression of the central spinal canal and lateral recess. The lateralized interlaminar approach medial facet joint resection was typically limited to the inferior articular process (IAP) by accessing the joint through the joint space. Typically, the interlaminar window is small or absent in the thoracic spine, extensive removal of the lamina and the lateral facet joint may be necessary using a high-speed burr. The transforaminal endoscopy was performed in the remaining 2 patients according to published protocols for the thoracolumbar spine [6,24,36,37,38].

For either technique, the patient is positioned (prone) and prepped in standard surgical fashion for surgery. Anatomic landmarks, such as the midline, the interlaminar window, the intervertebral disc space, and the facet joints, are marked on intraoperative PA and lateral views. Once the attack angles and the entry points are established, considering the location, size, and relationship to other anatomical structures, a small stab incision is made, typically 3 to 5 cm of the midline aiming for the facet joint with the interlaminar technique and 6 to 7 cm lateral for the transforaminal approach. All subsequent procedural steps are performed under lateral fluoroscopic guidance, maintaining parallel endplates, and most importantly, under direct and continuous videoendoscopic visualization. The intraoperative fluoroscopic identification of the thoracic surgical level is not always easy, with the wrong level surgery rate reported at 10 percent. If in doubt, the surgeons should consider an intraoperative consultation with the radiologist. The working cannula is inserted over sequential dilators positioning it at the medial aspect of the facet joint at its junction with the lamina. The interlaminar window is developed with a power drill. Once it is opened, the beveled side of the working cannula should be turned towards the ligament flavum. The inferior lateral edge of the lamina may be exposed with rongeurs, a radiofrequency probe, and burs as needed. As the decompression removes the inferomedial portion of the facet joint, the working cannula can be advanced into the lateral recess. The IAP resection should accommodate the endoscopic working cannula, typically 9 mm in outer diameter, to minimize medial retraction of the spinal cord. Complete resection of the thoracic facet joint is sometimes required on the approach side. The ligamentum flavum has to be opened with a blunt dissector and subsequently removed using endoscopic forceps and Kerrison rongeurs. The intervertebral disc should be scrutinized. Suppose the offending compressive pathology is not found. In that case, wrong-level surgery should be assumed. The entire process of identifying the symptomatic herniated disc should be repeated until the whole team is convinced that the correct thoracic level has been approached. After exposure of the herniated disc, adhesion with dura and nerve root should be separated with an elevator or dissector. After completing the decompression, the cannula will be turned to visualize the neural structures and control the decompression by a palpation hook.

### 2.6. Statistical and Outcome Analysis

Patients’ pain and recovery were monitored during the first 6–12 weeks following their treatment. The reduction of visual analog pain score (VAS) for back pain [39] was determined at final follow-up. Descriptive statistics were performed using IBM SPSS Statistics software, Version 27.0. The mean, range, standard deviation, and percentages of all nominal variables were calculated.

## 3. Results

### 3.1. Adamkiewicz Entry-Level

The most common surgical indication was unrelenting pain from a herniated nucleus pulposus (HNP) and foraminal stenosis. Table 1 lists the foraminal entry-level and laterality of the magna artery determined by CTA. The Adamkiewicz system was radiographically identified in 14/15 (93.33 %) patients (Table 1). In five of the fourteen patients with conclusive CTAs, the magna radicularis entered the spinal canal on the ventral surface of the exiting nerve root through the neuroforamen at the surgical level prompting a change in surgical strategy.

### 3.2. Patient Characteristics

The surgical level distribution and painful pathology encountered in our patients are listed in Table 2. Patients suffered from symptomatic thoracic herniated disc herniations from T4 to T12. Only the T8/9 and T12/L1 levels were treated twice. Each other level was treated once, suggesting that painful thoracic disc herniations have no particular predilection around the thoracolumbar junction. Ultimately, only six of the fifteen study patients underwent endoscopic decompression. The remaining nine patients were deemed inappropriate because of the proximity of the painful lesion to the AKA, the complexity and extent of the compressive pathology involving the central canal, or because of calcifications below the dural sac. Medical comorbidities were additional reasons. The surgical procedures performed and the problems encountered are listed in Table 3 and Table 4.

### 3.3. Relationship between Adamkiewicz and Painful Pathology

There were eight patients where the painful pathology was found far from the AKA foraminal entry-level (type 1), three patients with near location (type 2), and another four patient needing decompression at the foraminal (type 3) entry-level). In seven patients, the surgical level was either at the level or at the adjacent level to the foraminal entry level of the Adamkiewicz system. In five of the fifteen patients, the magna radicularis entered the spinal canal on the ventral surface of the exiting nerve root through the neuroforamen at the surgical level prompting a change in surgical strategy.

### 3.4. Choice of Treatment

The study patients’ clinical data regarding magna and surgical level, encountered painful pathology, provided treatment, and encountered problems are tabulated in Table 4. Patients with centrally located calcified large herniated discs and multilevel or severe foraminal stenosis were found unsuitable for endoscopic decompression. These patients had formal open decompression either with laminectomy or with laminectomy fusion as dictated by the underlying pathology. Patients with burst fractures or deformity and instability were also excluded from the endoscopic decompression protocol. Others for whom the surgery was deemed to be dangerous or unfeasible to be carried out open or endoscopically, underwent interventional care with transforaminal epidural steroid injections (TESI) for their painful herniated disc.

### 3.5. Clinical Outcomes

There were statistically significant reductions from the mean preoperative VAS of 8.53 ± 2.06 to the postoperative VAS of 1.60 ± 0.92 (*p* < 0.0001). The mean VAS reduction was 6.93 ± 1.86. Fourteen (93.33 %) of the fifteen patients had significant symptom resolution and reported better day-to-day function. One patient had incomplete neurological deficits consistent with anterior spinal cord syndrome after the endoscopic discectomy (patient #15). There were no cases of infection, dysesthesia, or numbness, but one patient with paralysis (patient #8; Table 4). In the latter patient, the AKA system was identified with CTA postoperatively. This complication prompted the authors to revise their preoperative work-up protocol for the thoracic herniated disc which included CTA from then on (Figure 2). 

In all of the remaining 14 study patients but one, the CTA study showed good dye filling of the intercostal vessels allowing us to trace it back to its origin at the descending aorta (Figure 3). 

## 4. Discussion

Endoscopic spine surgery is gaining more popularity beyond the lumbar spine. Employing this minimally invasive surgical technique in the thoracic spine may jeopardize the spinal cord vascular supply, which relies on three main arteries. The anterior spinal artery supplies the spinal cord’s anterior two-thirds versus the posterior one-third, provided by the two posterolateral spinal arteries [33,40]. The major contributory—the anterior spinal artery—originates from the two vertebral arteries at the level of the foramen magnum. It receives tributaries from anterior segmental medullary vessels at each spinal level from the aorta. The artery of Adamkiewicz is the biggest of these segmental medullary arteries. Alternative synonymous names used in the literature are arteria radicularis magna or the great anterior radiculomedullary artery [41]. Its course can be traced back to its starting from the descending aorta [42]. There, eight to ten segmental intercostal and lumbar arterial vessels divide into an anterior and posterior direction. These posterior tributaries arise from the radiculomedullary artery, the muscular, and the dorsal somatic branch. The more prominent component of the radiculomedullary artery splits off into the primary anterior and smaller posterior radiculomedullary arteries, and the largest anterior radiculomedullary artery is named the artery of Adamkiewicz. The artery of Adamkiewicz enters the intervertebral foramen adjacent to and often ventral to the exiting thoracic nerve in a slightly craniolateral direction traveling with the ventral root and ventral surface of the spinal cord [33]. It forms the classic “hairpin” arch seen on angiography when it joins the anterior spinal artery during its ascent [7,27,28].

There are anatomical variations of the AKA system. Typically, it arises from the left side of the aorta between T8 and L2 and most frequently between T9 to T12. However, it has been reported to enter the spinal canal as low as the L2/3. The AKA was found serendipitously after a patient developed spinal cord infarction after a right-sided transforaminal epidural steroid injection [43]. The authors recommended routinely injecting low in the neuroforamen just above the caudal pedicle [43]. In about 15% of patients, the AKA is found at T8 with a documented diameter ranging from 0.6 to 1.8 mm [7]. Additional anatomical variations include more than one AKA or the AKA system arising from the aorta on the right side or outside the typical T8 through L2 range. Additionally, the angle of how the AKA joints into the anterior spinal artery system may vary [7]. At times, collateral circulation from the muscular, intercostal, or lumbar arteries may be found, particularly if the AKA is occluded [29]. The consequence of injury to the AKA system is an anterior spinal cord syndrome with motor deficits and typically preserved sensory function [40,42]. Neurologic damage may manifest as fecal and urinary incontinence.

The authors’ preference in endoscopic thoracic spine surgery is the posterior interlaminar approach through the facet joint. Only two cases were performed via the transforaminal approach. The interlaminar window must be widely opened with a drill to visualize the dural sac and the exiting nerve root. The decompression evolves into a subtotal or complete facetectomy, beginning with the IAP in many cases. No iatrogenic instability was observed in our case series, which is likely due to the inherent stability of the thoracic rib cage and spine. A recent publication by Ruetten et al. on the transthoracic lateral retropleural approach advocates coagulation, or ligature of the intercostal artery [6]. However, these authors did not specifically advocate preoperative identification of the Magna radicular artery in their article or any of their others on full-endoscopic thoracic spine surgery [5,6,44].

In contrast, this team of authors recommends a formal CTA study to visualize the Adamkiewicz system and confirm its entry-level into the thoracic spinal canal. This change in the preoperative diagnostic workup was prompted by one patient who developed paraplegia postoperatively—a catastrophic event for the patient and members of the patient’s family and medical team. The radiographic CTA protocols employed in this study have been published and are commonplace [26,32]. The value of MRI angiography in comparison to CTA in the study of the AKA system is currently being evaluated [27]. Since the AKA may not be present in up to 10 percent of patients, [8] the radiologist must identify in the arterial study the intercostal arteries in their entirety, which typically also is associated with filling of the radicular arteries with contrast dye [26]. It may not be possible to say with certainty if an AKA is present or not if a deviation from this interventional radiology protocol occurred. The authors’ call to attention is not intended just for endoscopic spinal surgery in the thoracic spine but for all surgical interventions commonly performed for infection, tumor, fracture, and deformity.

Based on the authors’ small clinical series, we are proposing to stratify thoracic discectomy patients into three groups depending on the relationship between the painful pathology and the AKA. In the far-Type, the pathology is not close to the AKA as commonly seen with a herniated disc, tumor, and vertebral fracture. Endoscopic decompression can be considered. In the near-Type, the pathology is not located in the AKA thoracic entry foramen but in close proximity to the AKA, for example, at an adjacent level. The choice of preferred surgical technique and approach may be controversial. If surgery is indicated, the preoperative surgical planning must include delineating procedural steps to deal with the AKA when relieving spinal cord compression. When mobilization and retraction of the spinal cord are necessary, it should be delicately handled with minimal retraction for a brief moment since the clinical syndrome prompting the decompression is likely already caused by ischemia. As illustrated by the authors’ retrospective consecutive case series, this type of patient may be better off with a wide laminectomy or more extensive costotransvesectomy rather than endoscopic surgery, particularly if one is in doubt about being able to attack this type of problem endoscopically to avoid excessive spinal cord manipulation and additional ischemic insult. In the foraminal type, the compressive pathology is located precisely at the entrance level of the Magna root artery. Neither the nerve root nor the Magna artery must be “ligated” or coagulated regardless of the surgical technique. Temporary clipping of the intraforaminal magna radicularis with an aneurysm clip could be considered in open cases or whenever technically feasible, particularly when the CTA was non-conclusive. Low-flow vascular shunts fed by radicular arteries in patients with myelopathy have been temporarily clipped during the definitive treatment of craniospinal arteriovenous malformations and/or fistulas [45]. If there is no change in neurological monitoring with either somatosensory evoked potential (SSEPs) or motor evoked potentials (MEPs) or no heaviness or weakness in the lower extremities in awake patients, coagulation or ligation of the radicular artery could be considered. If there is transient weakness or dysesthesias in the lower extremities following temporary clipping, then the clip can be removed, and the blood pressure increased to lower the risk of neurological deficit. The neurosurgical literature recommends this protocol for Type I spinal dural arteriovenous fistulas (SDAVFs) and tumors [45]. Theoretically, it could have been performed in one of our patients with an AV fistula. However, this team of authors recommends a formal CTA as a more practical method since not every endoscopic spinal surgeon may have a neurosurgical training background.

Casual coagulation during the transforaminal access to the neuroforamen with the overzealous use of radiofrequency to control bleeding and improve visualization is often performed. Since the result could be catastrophic, the authors generally prefer the slightly lateralized interlaminar approach as a workhorse endoscopic technique with a direct trajectory to the facet joint space. However, the transforaminal approach may be more suitable in patients with overt myelopathy due to a centrally located and calcified hernia with associated myelomalacia on the thoracic MRI scan. These calcified central herniations represent the biggest challenge in endoscopy of the thoracic spine. The transforaminal approach is less likely to manipulate or retract the affected spinal cord since its working trajectory aims below the dural sac into the intervertebral disc. It could be considered a reasonable alternative to the interlaminar approach, mainly if the location of the AKA system has been established via CTA and random coagulation of this foraminal root artery is not performed.

Preoperative planning needs to carefully evaluate whether the patients’ compressive pathology can be reached with the standard access trajectory of the interlaminar or transforaminal approach. Far-migrated herniations or bony pathology distant to the thoracic surgical intervertebral disc space should perhaps be treated with more traditional open surgeries where broad exposure of the sensitive spinal cord can be easily accomplished. Nonetheless, the choice of the most suitable surgical approach must be made in conjunction with the surgeons’ skill level and experience with the endoscopic procedure.

## 5. Conclusions

The authors’ study illustrated that the artery of the Adamkiewicz system might be similarly at risk during endoscopic as during open thoracic spine surgery. The level of its foraminal entry location into the thoracolumbar spine should be determined before surgery by tracing the AKA system back to its origin at the descending aorta. Case reports even suggest that it can also be found in the mid-lumbar spine where it can become problematic for spinal injections. Our limited observational case series clinical results with the endoscopic decompression are favorable. However, this less burdensome surgery does have the same potential as open thoracic spine surgery to cause harm if the arterial blood supply to the thoracic spinal cord is compromised. Surgeons should be aware of this potentially catastrophic complication and not play Russian Roulette. They need to understand the surgical anatomy of their patients well before taking them to the operating room.

## Figures and Tables

**Figure 1 jpm-13-00356-f001:**
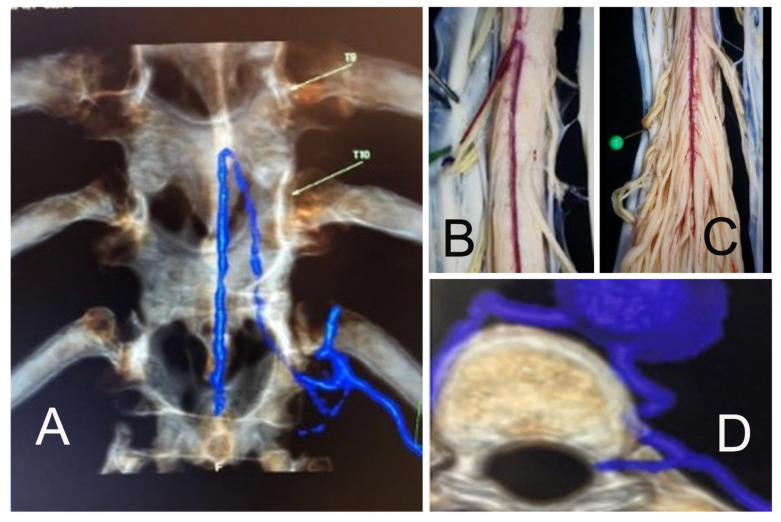
Computed Tomography (CT) angiography shows the entry foramen of the magna radicular artery or “Adamkiewicz” system in both coronal (**A**) and axial (**D**) cuts. A dissection of a cadaver spinal cord after wide longitudinal durotomy shows how the Magna Adamkiewicz system arises from a radicular artery (**B**) and supplies the majority of the thoracolumbar spinal cord and cauda equina (**C**).

**Figure 2 jpm-13-00356-f002:**
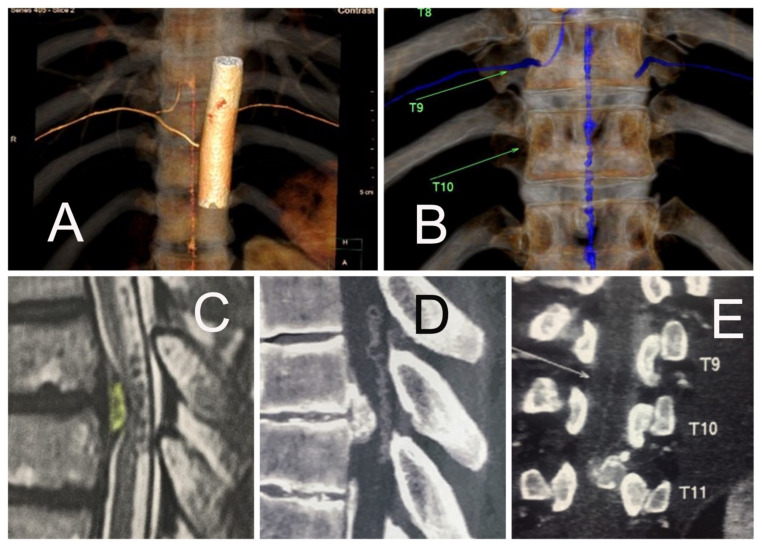
Various coronal CT angiography scans are shown (**A**,**B**) demonstrating the foraminal entry-level of the magna radicular artery at the T9-T10 and a medullary angioma, (**B**) at the same level of the thoracic disc herniation, (**C**–**E**) in the same patient at the T10-T11 level. The relationship between the thoracic surgical level and the foraminal entry-level of the blood supply to the Adamkiewicz system should be studied preoperatively before decompression. The causal use of electrocautery to improve visualization or access to the compressive pathology should be avoided. Unintended spinal cord ischemia during the endoscopic operation of the thoracic spine may result in postoperative neurological deficits. In this case, the surgeon decided to cancel the transforaminal endoscopic discectomy procedure because of the heightened risk of spinal cord ischemia.

**Figure 3 jpm-13-00356-f003:**
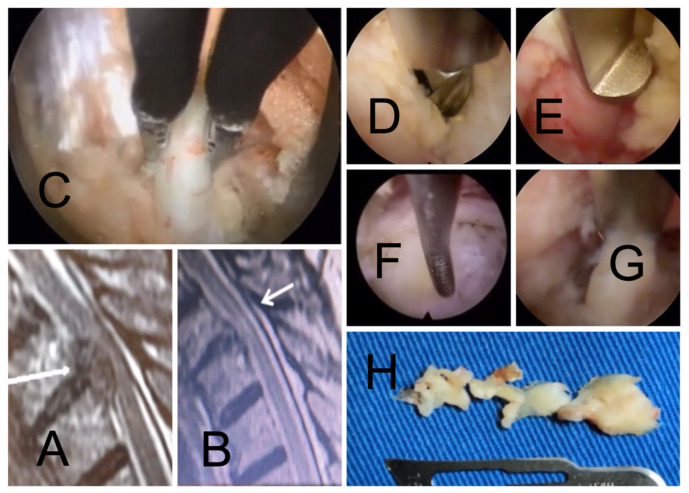
The preoperative (**A**) and postoperative (**B**) sagittal T2-weighted MRI scans of a patient who underwent transforaminal endoscopic T10/11 discectomy are shown. The white arrows point to the compressive pathology. The first author encountered the magna radicular artery (**C**) preventing direct access to the surgical neuroforamen. Instead of cauterizing and transecting it, the surgeon performed a wide foraminoplasty to improve access to the herniated disc with an endoscopic drill (**D**) and Kerrison rongeur (**E**). These tactical maneuvers allowed mobilization, retraction, and preserving this important contributor to the Adamkiewicz vascular supply system of the thoracolumbar spinal cord and cauda equina. The discectomy was completed by delivering the herniation from below the posterior longitudinal ligament with an endoscopic nerve hook (**F**) and removing it with an endoscopic grasper (**G**) piecemeal (**H**).

**Table 1 jpm-13-00356-t001:** Foraminal entry-level and side of the Artery of Adamkiewicz system.

Magna Level	Frequency	Percent	Cumulative Percent
L2/3	1	6.7	6.7
NOT IDENTIFIED	1	6.7	13.3
T10/11	3	20.0	33.3
T11/12	3	20.0	53.3
T12/L1	1	6.7	60.0
T7/8	1	6.7	66.7
T8/9	1	6.7	73.3
	4	26.7	100.0
Total	15	100.0	

SIDE	Frequency	Percent	Cumulative Percent
LEFT	7	46.7	46.7
NOT IDENTIFIED	1	6.7	6.7
RIGHT	7	46.7	46.7
Total	15	100.0	100.0

**Table 2 jpm-13-00356-t002:** Surgical level and symptomatic pathology.

Surgical Level	Frequency	Percent	Cumulative Percent
T10/11 RIGHT	1	6.7	6.7
T11/12 RIGHT	1	6.7	13.3
T12/L1 CENTRAL	2	13.3	26.7
T4/5 LEFT	1	6.7	33.3
T5/6 LEFT	1	6.7	40.0
T5/6 RIGHT	1	6.7	46.7
T6/7 CENTRAL	1	6.7	53.3
T6/7 RIGHT	1	6.7	60.0
T7/8 LEFT	2	13.3	73.3
T7/8 RIGHT	1	6.7	80.0
T8/9 RIGHT	2	13.3	93.3
T9/10 LEFT	1	6.7	100.0
Total	15	100.0	

PATHOLOGY	Frequency	Percent	Cumulative Percent
BONY FORAMINAL STENOSIS	3	20.0	20.0
BURST FRACTURE WITH CORD COMPRESSION	1	6.7	26.7
CALCIFIED HNP	3	20.0	46.7
HNP	7	46.7	93.3
RETROLISTHESIS, DEFORMITY, NEUROFIBROMATOSIS	1	6.7	100.0
Total	15	100.0	

HNP—Herniated nucleus pulposus.

**Table 3 jpm-13-00356-t003:** Surgical treatment and encountered problems.

Treatment	Frequency	Percent	Cumulative Percent
COSTOTRANSVERSECTOMY	1	6.7	6.7
ENDOSCOPIC DECOMPRESSION	6	40.0	46.7
LAMINECTOMY AND FORAMINOTOMY	1	6.7	53.3
LAMINECTOMY AND FUSION	2	13.3	66.7
NON-SURGICAL/TESI	3	20.0	86.7
NON-SURGICAL TREATMENT	2	13.3	100.0
Total	15	100.0	

PROBLEM	Frequency	Percent	Cumulative Percent
N.A.	8	53.3	53.3
AV FISTULA	1	6.7	60.0
LARGE CALCIFIED CENTRAL HNP	2	13.3	73.3
MULTILEVEL SEVERE STENOSIS	1	6.7	80.0
NEUROFIBROMATOSIS, RETROLISTHESIS AND DEFORMITY	1	6.7	86.7
POSTTRAUMATIC INSTABILITY	1	6.7	93.3
UNABLE TO FIND MAGNA	1	6.7	100.0
Total	15	100.0	

TESI—Transforaminal epidural steroid injection; HNP—Herniated nucleus pulposus.

**Table 4 jpm-13-00356-t004:** Clinical data of study patients.

No	Magna Level	Magna Type	Magna Side	Surgical Level And Side	Pathology	Treatment	Problem	Complication
1	T9/10	1	RIGHT	T8/9 RIGHT	HNP	ENDOSCOPIC DECOMPRESSION		
2	T8/9	1	LEFT	T5/6 RIGHT	BONY FORAMINAL STENOSIS	ENDOSCOPIC DECOMPRESSION		
3	T11/12	2	RIGHT	T6/7 CENTRAL	CALCIFIED HNP	NON-SURGICAL TREATMENT	AV FISTULA	
4	L2/3	1	LEFT	T6/7 RIGHT	HNP	ENDOSCOPIC DECOMPRESSION		
5	T9/10	1	LEFT	T5/6 LEFT	HNP	NON-SURGICAL/TESI		
6	T10/11	1	RIGHT	T4/5 LEFT	HNP	NON-SURGICAL/TESI		
7	T11/12	3	LEFT	T12/L1 CENTRAL	RETROLISTHESIS, DEFORMITY	LAMINECTOMY AND FUSION	NEUROFIBROMATOSIS, RETROLISTHESIS & DEFORMITY	
8	T9/10	2	LEFT	T7/8 LEFT	CALCIFIED HNP	COSTOTRANS-VERSECTOMY	LARGE CALCIFIED CENTRAL HNP	PARAPLEGIA
9	T11/12	3	RIGHT	T11/12 RIGHT	CALCIFIED HNP	NON-SURGICAL/TESI	LARGE CALCIFIED CENTRAL HNP	
10	NOT IDENTIFIED	1	NOT IDENTIFIED	T7/8 LEFT	HNP	ENDOSCOPIC DECOMPRESSION	UNABLE TO FIND MAGNA	
11	T9/10	1	LEFT	T9/10 LEFT	BONY FORAMINAL STENOSIS	NON-SURGICAL TREATMENT	MULTILEVEL SEVERE STENOSIS	
12	T12/L1	1	LEFT	T8/9 RIGHT	BONY FORAMINAL STENOSIS	LAMINECTOMY AND FORAMINOTOMY		
13	T10/11	2	RIGHT	T12/L1 CENTRAL	BURST FRACTURE WITH CORD COMPRESSION	LAMINECTOMY AND FUSION	POSTTRAUMATIC INSTABILITY	
14	T10/11	3	RIGHT	T10/11 CENTRAL	HNP	ENDOSCOPIC DECOMPRESSION		
15	T7/8	3	RIGHT	T7/8 RIGHT	HNP	ENDOSCOPIC DECOMPRESSION		anterior spinal cord syndrome

TESI—Transforaminal epidural steroid injection; HNP—Herniated nucleus pulposus.

## Data Availability

The data presented in this study are available on request from the corresponding author.

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
