# Peer review of "Identification of the Magna Radicular Artery Entry Foramen and Adamkiewicz System: Patient Selection for Open versus Full-Endoscopic Thoracic Spinal Decompression Surgery"

_jpm, 2023, doi:10.3390/jpm13020356_

Round 1

Reviewer 1 Report

The article entitled “IDENTIFICATION OF THE MAGNA RADICULAR ARTERY ENTRY FORAMEN AND ADAMKIEWICZ SYSTEM: PATIENT SELECTION FOR OPEN VERSUS FULL-ENDO-SCOPIC THORACIC SPINAL DECOMPRESSION SURGERY” emphasize on this fact that the Adamkiewicz system might be similarly at risk during endoscopic as during open thoracic spine surgery.  

There are minor items that must be considered:

1- The methodology is precise and detailed. However, caption of figure 1 must be revised since the panel C and D are not addressed vividly in the caption.

2- The English format of the Informed Consent Statement should be uploaded as supplementary material.

3- The Ethical/trial Registration information is not addressed in the methods.

Author Response

We thank this reviewer for his thorough examination of our manuscript and favorable assessment.

Responses:

  1. As requested, we revised the caption of figure 1 to include panels C and D in the description. We also proofread the entire manuscript and corrected minor orthographical and grammatical errors.
  2. The English format of the Informed Consent Statement was uploaded as supplementary material.
  3. The ethical/trial registration information is now listed in the methods at the end of section 2.1 Study Design, which says: "The study's IRB approval number is CEIFUS 106-19."

Reviewer 2 Report

This paper performed a retrospective observational cohort study on patients scheduled for decompression of symptomatic thoracic herniated discs and spinal stenosis who underwent a preoperative computed tomography angiography (CTA) to assess the surgical risks by anatomically defining the foraminal entry level of the magna radicularis artery into the thoracic spinal cord and its relationship to the surgical level. Through scientific experimental design and detailed research results, the authors recommend stratifying patients according to the proximity of the magna radicularis artery to the compressive pathology with CTA to assess the surgical risk with targeted thoracic discectomy methods. Considering a detailed study has been carried out, I have only the below minor comments addressing paper writing, and once these are addressed the work is recommended for publication.

1)     Methods section: Experimental procedures should be better defined, and more information should be provided about the participants’ characteristics.

2)     It suggested that sent the manuscript to an English editing company for English proofreading. I hope that the level of English can be significantly improved in the revised manuscript.

Author Response

We thank this reviewer for his detailed examination of our manuscript and favorable evaluation.

Responses:

  1. We added a section 2.2. Experimental Procedures at the beginning of the Method section to comply with this reviewers request to better define the goals of our study. We provided more information about the participants’ characteristics. All changes are highlighted in yellow.
  2. We submitted the manuscript to a strenuous proofreading process and several minor orthographical and grammatical improvements. We hope that this reviewer finds the level of English significantly improved enough to recommend the publication of our revised manuscript.